Ecological indicators reveal historical regime shifts in the Black Sea ecosystem

http://orcid.org/0000-0002-2814-3527 Akoglu Ekin eakoglu@metu.edu.tr
Institute of Marine Sciences, Middle East Technical University , Erdemli, Mersin , Türkiye
Semprucci Federica
Electronic publication date: 2023 Jul 11
Publication date: 2023
Volume: 11
Electronic Location ID: e15649
Received 2023 Mar 21; Accepted 2023 Jun 6
Copyright: © 2023 Akoglu
Copyright year: 2023
Copyright holder: Akoglu
License: This is an open access article distributed under the terms of the Creative Commons Attribution License, which permits unrestricted use, distribution, reproduction and adaptation in any medium and for any purpose provided that it is properly attributed. For attribution, the original author(s), title, publication source (PeerJ) and either DOI or URL of the article must be cited.
License URL: https://creativecommons.org/licenses/by/4.0/

Keywords: Ecological modelling, Regime shifts, Ecopath with Ecosim, Black Sea, Fisheries, Marine food webs

Funding: The author received no funding for this work.

==============================
Background

The Black Sea is one of the most anthropogenically disturbed marine ecosystems in the world because of introduced species, fisheries overexploitation, nutrient enrichment via pollution through river discharge, and the impacts of climate change. It has undergone significant ecosystem transformations since the 1960s. The infamous anchovy and alien warty comb jelly Mnemiopsis leidyi shift that occurred in 1989 is the most well-known example of the drastic extent of anthropogenic disturbance in the Black Sea. Although a vast body of literature exists on the Black Sea ecosystem, a holistic look at the multidecadal changes in the Black Sea ecosystem using an ecosystem- and ecology-based approach is still lacking. Hence, this work is dedicated to filling this gap.

Methods

First, a dynamic food web model of the Black Sea extending from 1960 to 1999 was established and validated against time-series data. Next, an ecological network analysis was performed to calculate the time series of synthetic ecological indicators, and a regime shift analysis was performed on the time series of indicators.

Results

The model successfully replicated the regime shifts observed in the Black Sea. The results showed that the Black Sea ecosystem experienced four regime shifts and was reorganized due to effects instigated by overfishing in the 1960s, eutrophication and establishment of trophic dead-end organisms in the 1970s, and overfishing and intensifying interspecies trophic competition by the overpopulation of some r-selected organisms (i.e., jellyfish species) in the 1980s. Overall, these changes acted concomitantly to erode the structure and function of the ecosystem by manipulating the food web to reorganize itself through the introduction and selective removal of organisms and eutrophication. Basin-wide, cross-national management efforts, especially with regard to pollution and fisheries, could have prevented the undesirable changes observed in the Black Sea ecosystem and should be immediately employed for management practices in the basin to prevent such drastic ecosystem fluctuations in the future.

Introduction

Globally, 35.6% of fish stocks in marine ecosystems are overfished, 57.3% are maximally sustainably fished in 2019, and the number of stocks fished at biologically sustainable levels is decreasing (FAO, 2022). Overfishing is the main instigator of ecological extinctions in coastal ecosystems (Jackson et al., 2001) and climate change is predicted to amplify biomass declines (Lotze et al., 2019). Large marine ecosystems (LMEs), which have distinctive bathymetric and hydrographic characteristics (Sherman & Duda, 1999) and host large fish populations with trophic (prey-predator) interactions (Alexander, 1993), have also been impacted by overfishing (Coll et al., 2008; Link et al., 2020; Link, 2021). As of 2019, 63.4% of fish stocks were fished at unsustainable levels in the Mediterranean and Black Sea large marine ecosystems (FAO, 2022). The Black Sea, one of the sixty-four large marine ecosystems in the world (Hempel & Sherman, 2003), has experienced multiple stressors from considerable fisheries overexploitation (Mee, 1992) to almost all of the documented anthropogenic stressors, such as marine litter (Topçu & Öztürk, 2010; Ioakeimidis et al., 2014); pollution and related nutrient enrichment, eutrophication and hypoxia (Pokazeev, Sovga & Chaplina, 2021); coastal abrasion and erosion (Kosyan & Velikova, 2016; Tătui et al., 2019) and the introduction of alien invasive species (Kideys, 2002; Shalovenkov, 2019) since the 1960s.

The Black Sea ecosystem has undergone a series of transformations since 1960 due to combinations of multiple stressors acting concomitantly and/or sequentially. In the 1960s, the Black Sea was in a quasi-pristine state characterized by high benthic and pelagic biodiversity, and its food web was dominated by apex predators such as Atlantic mackerel, bluefish, and Atlantic bonito, which exerted top-down (predator) control in the food web. With the onset of the 1970s, intense nutrient enrichment through rivers and overexploitation of apex predators during the previous decade caused the ecosystem shift paradigm to become bottom-up (resource) controlled (Oguz & Gilbert, 2007). During the 1970s and the 1980s, the food web of the Black Sea was dominated by small pelagic fish, mainly anchovy, until the infamous collapse of the anchovy stock and the outburst of the alien warty comb jelly, Mnemiopsis leidyi (Agassiz, 1865), in 1989 (Kideys, 2002). Fisheries yield, which reached 750 kilotonnes in the 1980s, abruptly declined to 200 kilotonnes in 1989 (Oguz, Akoglu & Salihoglu, 2012). In the 1990s, the Black Sea ecosystem transitioned to a mesotrophic stage with alleviating eutrophic conditions due to protective measures (Dorofeev, 2009), and moderate levels of fisheries yield have been realized ever since (Oguz, Akoglu & Salihoglu, 2012).

Overfishing could be considered the major instigator of changes in the Black Sea; however, all of the aforementioned stressors deteriorated its ecosystem and biodiversity (Bakan & Büyükgüngör, 2000). This combination of multiple stressors acting in one large marine ecosystem and their related impacts make the Black Sea a natural laboratory to investigate ecosystem-wide changes of anthropogenic disturbances (Konovalov et al., 2006) and necessitate an ecosystem-based approach. Ecological and statistical models are useful tools for studying ecosystems and their changes holistically and for improving our understanding of disturbances in marine environments. Several authors have employed ecological and statistical models and analysis of long-term time-series data to investigate historical changes in the Black Sea. The collapse of anchovy stocks and the outburst of M. leidyi in 1989 have been attributed to overfishing (Gucu, 2002) or trophic cascades triggered by overfishing (Daskalov, 2002; Daskalov et al., 2007). Based on the statistical models of time series of environmental variables and biomasses and catches of fish species, the multidecadal changes in the Black Sea were mainly attributed to the decreased resilience of the ecosystem by the removal of its apex predators which had a stabilizing role in the ecosystem through fisheries (Llope et al., 2011; Daskalov et al., 2017). However, a more holistic assessment based on population dynamics and mass-balance models indicated that climate-induced forcings, increased resource competition with alien invasive and indigenous opportunistic species, and trophic cascades exerted by overfishing had concomitant and sometimes sequential impacts on the multidecadal changes observed in the ecosystem (Oguz, Salihoglu & Fach, 2008; Akoglu et al., 2014). Thus far, research efforts have capitalized on linear statistical models that quantified the effects and relationships between the drivers and the related changes but provided limited understanding (Llope et al., 2011; Daskalov et al., 2017), a pelagic pooled ecosystem model that represented only a specific part (e.g., lower-trophic-level) of the food web and was coupled to a single-species population dynamics model (Oguz, Salihoglu & Fach, 2008), static aggregate models that represented average conditions in different decades (Örek, 2000; Gucu, 2002; Akoglu et al., 2014), or a temporal whole-of-ecosystem model that was not validated with time-series data (Daskalov, 2002). The only validated temporal model of the historical Black Sea focused on the infamous anchovy-Mnemiopsis shift in 1989 and investigated the environmental consequences of the introduction of M. leidyi (Berdnikov et al., 1999). However, it lacked an ecosystem-based assessment, and the model validation was limited to small pelagic fish stocks. A modelling study also focused on the Black Sea ecosystem’s status since the 2000s for developing fisheries management advice (Salihoglu et al., 2017). However, an assessment using ecological indicators that builds on the fundamentals of ecology sensu Odum (1969) to aid in the future ecosystem-based management approach by improving the understanding of these historical changes in the Black Sea is still missing, and a temporal whole-of-ecosystem model that focuses on the history of the Black Sea ecosystem and is validated against long-term time-series data is yet to be implemented.

Our study is the first whole-of-ecosystem time-dynamic model of the Black Sea that was validated against 40 years of available time series data for investigating multidecadal historical changes in the Black Sea ecosystem by employing ecological network analysis. We set up an Ecopath with Ecosim model to represent the quasi-pristine stage (the 1960s) of the Black Sea ecosystem and simulated multidecadal changes until 2000. Dynamical changes in the food web were investigated using time series of synthetic ecological indicators provided by the network analysis. We targeted to develop a holistic ecosystem-based understanding of historical changes in the Black Sea using the fundamental concepts of ecology sensu Odum (1969) and Ulanowicz (2000). Specifically, our aim was to: (i) develop a model of the Black Sea ecosystem representing the period from the quasi-pristine stage of the Black Sea (the 1960s) until the end of the 20th century and validate it against time series of statistical and field data until 2000, (ii) delineate the ecosystem structure and function in the Black Sea in the 1960s using the validated model, and finally (iii) understand the dynamical changes in the Black Sea using a series of ecological indicators that quantify the impacts of different drivers acting on the system and determine different regimes prevailed by considering the 1960s as a reference period.

Materials and Methods

Study area

The Black Sea is the largest semi-enclosed meromictic basin in the world with a permanent anoxic layer below 100–150 m (Sabatino et al., 2020, Fig. 1). The anoxic layer is characterized by significant H2S concentrations; therefore, benthic life is strictly confined to continental shelf regions. The upper layer, from the surface down to 50 m, is well-oxygenated, and a suboxic layer with oxygen concentrations below 10 µM lies between these two layers (Murray, Top & Özsoy, 1991). Redox reactions occur in the suboxic-anoxic interface zone, which significantly influences the biogeochemistry of the Black Sea (Murray et al., 2007). The only connection of the Black Sea is the narrow Bosphorus Strait, through which Mediterranean waters enter the basin at depth, and Black Sea waters flow to the Marmara Sea at the surface. The catchment area of the basin is approximately 2 million km2, and more than 340 km3 of freshwater is carried into the Black Sea by major rivers, such as the Danube, Dnieper, Dniester, Bug, Sakarya, Carsamba, Kizilirmak, Yesilirmak, and Coruh, and their tributaries (Vespremeanu & Golumbeanu, 2018).

Figure 1 Modelling area: the Black Sea excluding the Azov Sea.

The Black Sea is surrounded by six countries: Bulgaria, Romania, Ukraine, Russia, Georgia, and Turkey. Significant anthropogenic activities take place in the basin; therefore, multiple stressors, including, but not limited to, eutrophication due to pollution, overfishing, the introduction of non-indigenous species (NIS), coastal erosion due to construction, and climate change, have acted on the Black Sea ecosystem since the 1960s. Furthermore, the catchment area of the Black Sea includes territories of 23 countries (Myroshnychenko et al., 2015), making it a perfect natural laboratory for studying anthropogenic impacts on marine ecosystems.

The modelling approach

One of the most widely used food web models, Ecopath with Ecosim (EwE, Christensen, Walters & Pauly, 2005) version 6.6.8, was used to set up a temporal model of the Black Sea between 1960–1999. First, Ecopath, the mass-balanced module of the EwE suite, was used to set up a model to represent the ecosystem conditions of the early 1960s in the Black Sea. Next, Ecosim, the temporal module of the EwE suite, was used to simulate the changes in the Black Sea ecosystem until 2000.

The mass-balance module Ecopath capitalizes on two master equations. The first equation ensures mass-balance as

Bi∗(PB)i−∑j=1n⁡Bj∗(QB)j∗DCji−(1−EEi)∗Bi∗(PB)i−Ei−Yi−BAi=0

where Bi is the biomass of group i per unit area, (P/B)i is the annual production to biomass ratio of group i, Bj is the biomass of predator j per unit area, (Q/B)j is the annual consumption to biomass ratio of predator j, DCji is the relative proportion of prey i in the diet of predator j, EEi is the ecotrophic efficiency, Ei is the annual net migration rate, Yi is the annual catch rate and BAi is the annual biomass accumulation rate of group i. The first multiplicative term calculates the production rate of group i, the second summation term calculates the total predation rate on group i, and the third multiplicative term is the other mortality rate of group i, which is due to old age, starvation, and/or diseases. Ecopath requires three of the four basic input parameters, Bi, (P/B)i, (Q/B)i, and EEi, and an additional relative diet composition matrix to be defined. The remaining missing parameter is calculated using the Ecopath algorithm.

The second master equation of Ecopath ensures the energy balance of groups as

Qi=Pi+Ri+Ei

where Qi, Pi, Ri, and Ei are the consumption, production, respiration, and egestion, respectively, of group i.

The Ecopath with Ecosim model of the Black Sea is based on Akoglu (2013); however, it was modified and updated with recently available literature data for the modelling period to improve the model’s representation of the Black Sea. The model includes 22 functional groups. Nine commercially important fish groups were included in the model: Atlantic bonito (Sarda sarda, Bloch, 1793), bluefish (Pomatomus saltatrix, Linnaeus, 1766), Atlantic mackerel (Scomber scombrus, Linnaeus, 1758), whiting (Merlangius merlangus, Linnaeus, 1758), turbot (Scophthalmus maximus, Linnaeus, 1758), red mullet (Mullus barbatus, Linnaeus, 1758), spiny dogfish (Squalus acanthias, Linnaeus, 1758), Mediterranean horse mackerel ((Trachurus mediterraneus, Steindachner, 1868), hereinafter referred to as “horse mackerel”), Pontic shad ((Alosa immaculata, Bennett, 1835), hereinafter referred to as “shad”), European sprat ((Sprattus sprattus, Linnaeus, 1758), hereinafter referred to as sprat), and European anchovy ((Engraulis encrasicolus, Linnaeus, 1758), hereinafter referred to as anchovy) as a multi-stanza group as adults (1,1+), and eggs and larvae (0+,1). One marine mammal group (dolphins) was included to represent the dolphins and porpoises of the Black Sea: common dolphin (Delphinus delphis, Linnaeus, 1758), bottlenose dolphin (Tursiops truncatus, Montagu, 1821), and harbor porpoise (Phocoena phocoena, Linnaeus, 1758). The model included one phytoplankton group as producer, one edible zooplankton group, one benthic invertebrate group, one heterotrophic dinoflagellate, Noctiluca scintillans ((Macartney) Kofoid & Swezy, 1921), that significantly grazes on edible zooplankton, and four jellyfish species: moon jelly Aurelia aurita (Linnaeus, 1758), the alien warty comb jelly Mnemiopsis leidyi (Agassiz, 1865), the alien brown comb jelly Beroe ovata (Bruguière, 1789), which is the predator of M. leidyi, and sea gooseberry Pleurobrachia pileus (Müller, 1776), being either trophic dead-end organisms, that is, not consumed by predators or having only specialized predators in the food web. Although the exact dates of the introductions of the alien species M. leidyi and B. ovata are uncertain, they did not exist in the Black Sea until the 1980s and the 1990s, respectively. In EwE models, it is not possible to introduce new state variables once the Ecopath model, which provides the initial conditions for the dynamic Ecosim model simulation, is set up. Therefore, these two introduced species were included in the 1960s’ Black Sea Ecopath model, but their biomasses were set to values close to zero and forced to be equal to zero until their periods of introduction.

For simplicity, only two fishing fleets, namely trawlers and purse seiners, were included in the model targeting demersal (red mullet, turbot, whiting, spiny dogfish) and pelagic (anchovy, sprat, shad, horse mackerel, bluefish, and bonito) fish groups, respectively. The marine mammals of the Black Sea were caught commercially until the ban on their fisheries in 1966 in the USSR, Bulgaria, and Romania, and in 1983 in Turkey (Birkun, 2008). Purse seining and shooting were the main methods for dolphin fisheries (Berkes, 1977); therefore, an amount of catch was defined to purse seiners fleet until 1983. The 1960s’ mass-balance model of the Black Sea was parameterized to represent the general food web conditions of the inner Black Sea basin to avoid extreme variability of the northwestern shelf (NWS). The model domain represents an area of 150,000 km2 where fisheries operate intensively in the vicinity of the EEZs of the riparian countries (Oguz, Salihoglu & Fach, 2008).

In accordance with the laws of thermodynamics and mass conservation, an Ecopath model is considered balanced when: (i) all EEs of the groups are less than unity and respirations of groups are non-negative (Christensen, Walters & Pauly, 2005); (ii) the production to consumption (P/Q) ratios, also known as gross food conversion efficiencies, are less than 0.5 (Piroddi et al., 2015); (iii) respiration to assimilation (R/A) and production to respiration ratios (P/R) of groups are lower than unity (Darwall et al., 2010); and (iv) respiration to biomass (R/B) ratios of groups are between 1–10 y−1 for fish and higher for lower-trophic-level groups (Heymans et al., 2016). However, R/B ratios can occasionally be greater than 10 y−1 for marine mammals (e.g., Blanchard, Pinnegar & Mackinson, 2002; Arreguín-Sánchez, Arcos & Chávez, 2002). We balanced the model to ensure mass and energy conservation and used pre-balance (PREBAL) diagnostics to increase the rigor of the final model and assess input data quality (Link, 2010). PREBAL diagnostics expect a linear positive slope from high to low trophic levels considering the Ecopath input parameters of B, P/B, Q/B and P/Q in line with general ecological principles.

Input parameters were collected from scholarly articles, technical reports, and books. For fish groups, Ecopath initial biomasses, catches, and production-to-biomass ratios, which are equal to total mortality (Z, y−1) under steady-state conditions (Allen, 1971), were obtained from published local studies. The consumption-to-biomass ratios of the fish groups were calculated using the empirical equation proposed by Palomares & Pauly (1998). For phytoplankton and zooplankton groups, biomasses were obtained from Nesterova et al. (2008) and Shiganova et al. (2008), respectively, and primary production estimates were obtained from Yunev (2011). The detritus biomass was calculated using the empirical equation of Pauly, Soriano-Bartz & Palomares (1993). Diet composition data were collected from the published literature, preferably for the Black Sea when available, and otherwise supplemented from adjacent ecosystems or using data for the same species from different ecosystems. The historical stomach content studies were limited in the Black Sea; therefore, we capitalized on literature information from all periods as well as recently available studies considering that the EwE model has a static diet specification in terms of prey items and does not take into account possible dietary changes with newly introduced or extinct species over time. The EwE model included three piscivorous fish species, namely Atlantic bonito, Atlantic mackerel and bluefish, which migrate to the Marmara Sea located in the southwest of the Black Sea for wintering and return back to the Black Sea in summer for reproduction. Therefore, to represent the feeding of these species when they were outside the model domain, an import fraction was defined, following Gucu (2002). The details and sources of the basic input parameters (B, P/B, Q/B, and EE) and the relative diet composition matrix (DC) are provided in Tables S1 and S2, respectively. The biomass and catch values in the literature were in tonnes and grammes of wet weights and converted to grammes carbon using conversion factors specific for each group/species in the model following Akoglu et al. (2014) and references therein.

The temporal Ecosim module of the EwE suite was used to simulate the changes in the Black Sea ecosystem from 1960 to 1999. For each functional group, Ecosim solves a set of ordinary differential equations of the form

dBidt=(PQ)i∑j=1n⁡Qji−∑j=1n⁡Qij+Ii−(Mi+Fi+ei)∗Bi

where dBi/dt is the rate of change in the biomass of group i, (P/Q)i is the gross food conversion efficiency of group i, Qji is the consumption of group i of its prey j, Qij is the predation of group i by its predator j, Ii is the immigration rate of i, Mi and Fi are the natural and fishing mortality rates, respectively, and ei is the emigration rate of group i. The consumption of groups, i.e., prey-predator interactions between groups, is formulated using the foraging arena approach (Ahrens, Walters & Christensen, 2012), which separates prey biomass into two compartments: vulnerable and invulnerable to predators. The exchange between these two compartments occurs instantaneously at every time step at a rate defined by the vulnerability parameter. This parameter can range from unity, a resource-driven (bottom-up) interaction, to infinity, a predator-driven (top-down) interaction. The default value of the vulnerability parameter in Ecosim is set to two in the model, indicating a mixed type of interaction.

Ecosim requires time series of primary productivity, fishing effort or mortalities to run a basic simulation, and time series of biomasses and catches are required for model validation. In this study, the time series of fishing mortalities obtained from Prodanov et al. (1997), Ivanov & Panayotova (2001), Shlyakhov & Daskalov (2008) and The Scientific, Technical and Economic Committee for Fisheries (STECF) report (STECF-15-16, 2015) was used to force the model’s fisheries catches. However, the fishing mortality time-series data for some species did not cover the entire modelling period. Therefore, the time series of fishing mortalities were prepended back to 1960 and/or appended until 1999 with exponential smoothing state space model (ETS) following Hyndman & Athanasopoulos (2018) in R (R Core Team, 2022) using package “forecast” (Hyndman & Khandakar, 2008). Considering the productivity of primary producers, decadal averages of primary productivity in the Black Sea from Yunev et al. (2002) were used to force the productivity of the phytoplankton group. The forcing function data for the temporal Ecosim model are given in Fig. S1.

A sequential fitting approach, as per Tomczak et al. (2012), was used to fit the model by alternating different sets of parameters, i.e., vulnerabilities and primary production anomalies, to the time series of stock assessment derived biomasses and catches from Prodanov et al. (1997), Ivanov & Panayotova (2001), Shlyakhov & Daskalov (2008) and The Scientific, Technical and Economic Committee for Fisheries (STECF) report (STECF-15-16, 2015) for fish groups, and time series of biomass values from Shiganova et al. (2008) and Nesterova et al. (2008) for fodder and gelatinous zooplankton, and phytoplankton groups, respectively. Considering the high uncertainty in the historical primary production estimates in the Black Sea, primary production anomalies were searched capitalizing on the decadal primary productivity estimates from Yunev et al. (2002). Five spline points were used to estimate a primary productivity anomaly curve based on the historical decadal changes in the primary productivity of the Black Sea. The time-dynamic Ecosim Black Sea model was assessed by considering the biomass and catch dynamics, and ecological function of the Black Sea ecosystem documented in the literature (Daskalov, 2002; Gucu, 2002; Oguz, Salihoglu & Fach, 2008; Llope et al., 2011; Akoglu et al., 2014; Daskalov et al., 2017). The model skill was analyzed by comparing the time series of simulated biomasses and catches, the stock assessment derived biomasses and statistical catches for fish stocks and available field-sampled data for other groups. The sum of squared deviations (SS) of log simulated data from log reference data (Christensen, Walters & Pauly, 2005), Akaike Information Criterion (Akaike, 1974) corrected for small sample sizes (AICc) and Taylor diagrams (Taylor, 2001), which show the correlations, normalized standard deviations, and root mean squared distances (RMSD) of simulated values from reference data, were used to assess the model skill. The best model was selected on the basis of the AICc value.

Indicator-based ecosystem assessment approach

The Ecopath model of the early 1960s was analyzed by capitalizing on ecological network analysis using a set of synthetic ecological indicators and ecosystem statistics. The maturity and developmental status of the Black Sea ecosystem were analyzed sensu Odum (1969) for the 1960s’ ecosystem.

The statistical properties of the flows in the food web and information indices were used to describe ecosystem characteristics. The total system throughput (TST), a macro-ecosystem indicator that is the sum of respiratory, consumption, export flows, and flows to detritus in the food web, was calculated to delineate the trophic size of the Black Sea food web. For instance, TST can increase with increasing productivity, e.g., eutrophication, in an ecosystem, indicating ecosystem growth (Christensen, 1995). Further, net system production (total primary production minus total respiratory flows) that is assumed to be close to zero in mature ecosystems (Odum, 1969), system omnivory index (SOI) that indicates the breadth of feeding interactions in the food web, relative primary production required to sustain fisheries catches (PPRc) that shows the fisheries’ ecological footprint on the ecosystem, mean trophic level of the catches (mTLc) and community (mTLco) that indicates the exploitation status of fish populations (Pauly et al., 1998), the ratio of total biomass to total system throughput that is assumed to be high in mature ecosystems, the ratio of total respiration to total biomass (R/B), also known as Schrödinger ratio that defines the developmental status of ecosystems and is assumed to be low in highly-structured mature ecosystems (Odum, 1969), and the ratio of total primary production to total respiration, which is expected to be close to unity for mature ecosystems, were calculated to assess the characteristics of the Black Sea food web. The mean transfer efficiency of flows, which is the geometric mean of the transfer efficiencies from trophic level (TL) II to the highest trophic level, was also calculated to assess the relative amount of energy passing through the different trophic levels (TLs) in the food web.

The indicators of relative throughput cycled in the food web, namely Finn’s cycling index (FCI), which shows the material cycling in the food web; predatory cycling index (PCI), which shows the material cycling in the food web excluding the detritus compartment; and Finn’s mean path length (PL), which is the average path length of a unit of energy travels (Finn, 1976), were calculated to assess the degree of material and energy cycling and the average number of TLs that a unit of energy travelled in the food web. These three indicators are expected to be high in resilient and healthy ecosystems (Odum, 1969).

System ascendency, overhead, and developmental capacity were calculated as information indicators to investigate the immunity status of the ecosystem against stress and its organizational structure (Costanza & Mageau, 1999; Ulanowicz, 2000). Ascendency shows the organizational degree in an ecosystem or food web, and overhead (resilience) is the strength of an ecosystem against stress. Akin to a healthy organism, a healthy ecosystem is expected to have a balanced degree of resilience to maintain its structure and function under stress and ascendency to recover from perturbed conditions, i.e., regime shifts.

Trophic flows in the food web were investigated using Lindeman spines (Lindeman, 1942) by integrating the flows into discrete TLs. Furthermore, mixed trophic impact analysis was used to assess the positive and negative interactions between the groups in the food web (Ulanowicz & Puccia, 1990). Mixed trophic impact (MTI) analysis can be considered similar to sensitivity analysis and indicates what would happen to other groups if a group’s biomass in the food web changed. The MTI index scales between negative one, indicating a strong negative impact, and positive one, indicating a strong positive impact. Finally, keystoneness analysis was used to determine the groups that have a structuring role in the food web dynamics. Keystone species or groups are organisms that have relatively low biomass in the system; however, they can have a huge impact on the food web dynamics if their biomasses change (Libralato, Christensen & Pauly, 2006).

Ecological network analysis of time series of ecological indicators

The time-dynamic Ecosim simulation of the Black Sea model was analyzed using time-series network analysis capitalizing on a set of ecological indicators produced by Ecosim Ecological Network Analysis (ENA). The time series of TST, relative ascendency (A/C), and relative overhead (O/C) were calculated as macro-ecosystem indicators. Indicators related to ecosystem function, i.e., PCI, FCI, PL, and the proportion of flows to detritus (PFD), which is an indicator of energy not utilized at higher trophic levels but diverted back to detritus, were calculated. The system primary production (PP), total system biomass, total catches, mTLc, and fishing in balance index (FiB), which is an indicator of the ecological balance of catches in terms of the trophic levels of the exploited species (Pauly & Watson, 2005), were calculated. An increase/decrease in the FiB index may indicate technological and geographical expansion/collapse of the fishery or the inclusion of new higher-trophic-level/lower-trophic-level species in the catches. If catches increase due to increased primary productivity, but the mTLc does not change or decrease, the FiB index increases. If catches decrease, the FiB index decreases. If a decrease in the mTLc is matched by a proportional increase in the catch, then FiB does not change. Furthermore, the loss in secondary production due to the fisheries exports index (L-index), relative PPRc, and the probability of the ecosystem’s being sustainably fished index (Psust) were calculated to quantify the impacts of fishery exploitation on the ecosystem. The ratios of predatory fish biomass to forage fish biomass, demersal fish biomass to pelagic fish biomass, and fish biomass to invertebrate biomass were calculated to delineate the impact of fisheries exploitation on the structure of the ecosystem, as all three indicators are expected to decrease with ongoing fisheries exploitation. Considering the diversity in the food web, Shannon’s diversity index, and the ratio of fodder zooplankton biomass to total zooplankton biomass, which can also be an important indicator of energy transfers in the food web, were calculated. First, the anomalies in the time series of the ecological indicators were calculated. Subsequently, possible regime shifts were investigated using Sequential T-test Analysis of Regime Shifts (STARS) analysis (Rodionov, 2004). Because of the autocorrelation in the time series of network indicators produced by the time-dynamic model simulation, a pre-whitening procedure was applied to remove red noise from the time series of ecological indicators’ anomalies using ordinary least-squares estimation for serial correlation. The Black Sea ecosystem is known to experience almost decadal shifts in its ecosystem structure (Oguz & Gilbert, 2007). Therefore, a cut-off length of 5 years and a significance level of p = 0.05 were used for regime shift detection, and regime shift indices (RSI) were estimated to detect abrupt changes, i.e., regime shifts, in the time series of indicators.

The time series of synthetic ecological indicators’ anomalies were also analyzed using principal component analysis (PCA) (Abdi & Williams, 2010). PCA is a linear ordination method used to analyze high-dimensional data and visualize the variation in a number of reduced dimensions. PCA helps delineate the most influential variables that significantly contribute to variability and identify variables that correlate together to explain the variability in the data. Using PCA, the time series of ecological indicators were also investigated based on historical decadal periods in the Black Sea. The number of principal components to retain was determined based on empirical Kaiser criterion (Braeken & van Assen, 2017) using “EFA.dimensions” package (O’Connor, 2023) in R (R Core Team, 2022).

Finally, the effects of fishing on the Black Sea ecosystem were analyzed using the time series of the L-index and calculated theoretical values of ecosystem-based maximum sustainable catches (EMSCp) for target L-index values (Libralato et al., 2008), that is, the 50% and 75% probabilities of the ecosystem being sustainably fished (Psust) and its corresponding L-index values, L50% and L75%. The temporal variability in the L-index was used to assess the historical changes in the Black Sea fisheries catch during the 40-year simulation period and the sustainability status of the fisheries in the Black Sea.

Results

The Black Sea model included 22 functional groups. Phytoplankton and detritus were the lowest trophic level groups (TL = 1), while turbot was the highest trophic level group with a trophic level of 4.64 (Table 1, Fig. 2). The predatory fish species included Atlantic bonito (TL = 4.16) and bluefish (TL = 4.14) as medium pelagic fishes, and whiting (TL = 3.97) and spiny dogfish (TL = 4.05) as demersal fish. The dolphins group had a trophic level of 4.22.

Table 1 Inputs (bold) and outputs of the Black Sea mass-balance Ecopath model by functional groups.

Group	Trophic level	Biomass (gC m−2)	P/B (y−1)	Q/B (y−1)	EE	P/Q	Catch (gC m−2 y−1)	
Seiners	Trawlers	
Dolphins	4.22	0.030	0.35	17.05	0.29	0.02	0.0002	–	
Atlantic bonito	4.16	0.059	0.82	7.55	0.97	0.11	0.0116	–	
Bluefish	4.14	0.015	0.96	5.11	0.99	0.19	0.0087	–	
Atlantic mackerel	3.04	0.020	1.43	8.66	0.92	0.16	0.0058	–	
Whiting	3.97	0.223	0.92	5.07	0.99	0.18	–	0.0030	
Turbot	4.64	0.016	0.55	2.76	0.03	0.20	–	0.0003	
Red mullet	3.09	0.013	1.66	7.25	0.97	0.23	–	0.0012	
Spiny dogfish	4.05	0.059	0.16	2.87	0.03	0.06	–	0.0003	
Horse mackerel	3.30	0.191	1.71	7.38	0.95	0.23	0.0131	–	
Shad	3.33	0.008	1.28	6.40	0.96	0.20	0.0011	–	
Sprat	3.04	0.412	1.70	10.58	0.98	0.16	0.0033	–	
Anchovy 1,1+	3.04	0.597	1.81	9.25	0.98	0.20	0.0151	–	
Anchovy 0,0+	2.83	0.001	6.44	73.07	0.00	0.09	–	–	
Benthic invertebrates	2.05	0.750	2.50	22.99	0.64	0.11	–	–	
A. aurita	2.80	0.048	10.95	34.76	0.00	0.32	–	–	
B. ovata	3.30	0.000	11.72	39.05	0.00	0.30	–	–	
M. leidyi	2.84	0.000	20.10	55.68	0.11	0.36	–	–	
P. pileus	2.05	0.010	10.95	29.20	0.00	0.38	–	–	
N. scintillans	2.04	0.110	7.30	36.20	0.94	0.20	–	–	
Zooplankton	2.04	0.400	38.00	152.00	0.94	0.25	–	–	
Phytoplankton	1.00	0.600	105.00	–	0.73	–	–	–	
Detritus	1.00	5.926	–	–	0.85	–	–	–	
Note:

P/B denotes production/biomass; Q/B denotes consumption/biomass; EE denotes ecotrophic efficiency, and P/Q denotes production/consumption.

Figure 2 Flow diagram of the 1960s’ Black Sea model.

Mass-balance model skill

The calculated pedigree index for the Black Sea mass-balance model was 0.49, indicating a moderate level of data quality considering the model inputs (Table 2). PREBAL diagnostics indicated that the biomasses of bluefish, shad, Atlantic mackerel, red mullet, A. aurita, P. pileus and N. scintillans could be underestimated, and the biomasses of whiting, horse mackerel, sprat, anchovy, benthic invertebrates and zooplankton could be overestimated (Fig. S2). The P/B values mostly conformed to a linear positive slope from high to low trophic levels; however, the P/B values of B. ovata, M. leidyi and phytoplankton could be overestimated. The Q/B values also conformed to the linear slope, with possibly overestimated values for dolphins, B. ovata, M. leidyi, the anchovy 0,0+ group and zooplankton. Because gross growth efficiencies (P/Q) were calculated using P/B and Q/B values, P/Q values of dolphins, spiny dogfish, anchovy 0,0+ and benthic invertebrates could be underestimated, and P/Q values of B. ovata, M. leidyi, A. aurita and P. pileus could be overestimated.

Table 2 Summary statistics for the 1960s’ Black Sea mass-balance Ecopath model.

Parameter	Value	Unit	
Sum of all consumption	98.1	gC m−2 y−1	
Sum of all exports	5.8	gC m−2 y−1	
Sum of all respiratory flows	57.48	gC m−2 y−1	
Sum of all flows into detritus	39.3	gC m−2 y−1	
Total system throughput (TST)	200.7	gC m−2 y−1	
Sum of all production	84	gC m−2 y−1	
Calculated total net primary production (TPP)	63	gC m−2 y−1	
Total primary production/total respiration	1.1	–	
Net system production	5.5	gC m−2 y−1	
Total primary production/total biomass	19.6	–	
Total biomass/total throughput	0.018	–	
Total biomass (excluding detritus)	3.56	gC m−2 y−1	
TE from primary producers	14.6	%	
TE from detritus	11.3	%	
TE total	13.4	%	
Mean TL of the community (mTLco)	1.58	–	
Mean TL of the community (TL ≥ 2)	2.77	–	
Marine Trophic Index (MTI) of community (TL ≥ 3.25)	3.81	–	
System omnivory index	0.15	–	
Shannon diversity index	2.24	–	
Pedigree index	0.49	–	
Total catch	0.064	gC m−2 y−1	
Mean TL of the catch (mTLc)	3.52	–	
Marine Trophic Index (MTI) of catch (TL ≥ 3.25)	3.83	–	
Gross efficiency (catch/net primary production)	0.001	–	
Loss of production index (L-index)	0.013	–	
Probability of sustainable fishing (Psust)	0.89	–	
PPRc to sustain the fishery/Total PP	5.8	%	
Throughput cycled (excluding detritus)	3.4	gC m−2 y−1	
Predatory cycling index (PCI)	2.8	%	
Throughput cycled (including detritus)	21.6	gC m−2 y−1	
Finn’s cycling index (FCI)	10.8	% of TST	
Finn’s mean path length (FPL)	3.2	–	
Ascendency (A)	26.5	%	
Overhead (O)	73.5	%	
Capacity (C)	795.4	Flowbits	

Mass-balance model of the early 1960s’ Black Sea

The TST of the Black Sea ecosystem in the early 1960s was 200.7 gC m−2 y−1 and consisted of 48.9% consumption, 2.9% export, 28.6% respiration flows, and 19.6% flows to detritus (Table 2). The net system production was 5.5 gC m−2 y−1, and the total primary production to total respiration ratio was 1.1. The total biomass supported per unit of energy (total biomass-to-total system throughput ratio) was 0.02 gC m−2 y−1. The SOI, calculated using the omnivory indices of the individual species/groups, was 0.15. The values of mTLc and PPRc were 3.52 and 5.8%, respectively. The gross efficiency of the system, that is, the ratio of catches to net primary production, was 0.001, indicating a relatively low level of fishery exploitation. However, a significant proportion of PPRc (>4%) consisted of catches from higher-trophic-level fish species in the ecosystem. The L-index was calculated as 0.012, indicating a high probability of sustainable fishing in the ecosystem. Considering the information indices, the system’s relative ascendency (A/C) and relative overhead (O/C) were 26.5% and 73.5%, respectively, indicating a highly resilient ecosystem. The details of the summary statistics of the system are presented in Table 2.

The relative amount of throughput cycled in the system was 21.6%, and the FCI and FPL were 10.8% and 3.2, respectively, indicating a high degree of cycling in the ecosystem. The PCI was calculated 2.8%. The mean transfer efficiency of the food web was 13.4%.

According to the Lindeman spine, the grazing food chain (from primary producers to higher trophic levels) dominated the flows in the food web as the total consumption flows originating from TL I (primary producers and detritus) to trophic level II consisted of 57.6% of flows from primary producers and 42.4% of flows from detritus (Fig. 3). The majority of exports occurred in TL III (0.037 gC m−2 y−1) and TL IV (0.025 gC m−2 y−1). The flows to detritus from TL II, III and IV were 18.4, 3.1 and 0.5 gC m−2 y−1, respectively.

Figure 3 Lindeman spine depicting trophic flows between integer trophic levels.

TL denotes trophic levels, P denotes primary producers, D denotes detritus compartment, TST denotes total system throughput and TE denotes transfer efficiency.

The mixed trophic impact analysis revealed strong direct negative impacts of trawlers on the demersal species turbot and spiny dogfish (Fig. 4). Seiners had strong direct negative impacts on piscivorous fish, namely Atlantic bonito, bluefish and Atlantic mackerel, and both direct and indirect negative impacts on dolphins due to hunting and fishing on prey species of dolphins, respectively. The heterotrophic dinoflagellate N. scintillans had a slightly negative impact on zooplankton. The majority of negative impacts on zooplankton were due to predation by anchovy and sprat. Zooplankton and phytoplankton positively impacted dolphins and all fish groups, except red mullet and benthic invertebrates due to indirect impacts that favored the predators of these two groups. Spiny dogfish negatively impacted dolphins due to direct predation and resource competition as they fed on similar species in the food web. Sprat positively impacted dolphins as a prey item; however, dolphins group was impacted negatively by adult anchovy group due to its strong positive impact on dolphins’ main competitor, i.e., spiny dogfish, in the food web. Dolphins group impacted all bony fish groups negatively due to direct prey-predator relationships; however, anchovy, red mullet and horse mackerel were the only exceptions because of the indirect positive impacts of dolphins on these species due to the consumption of their predators.

Figure 4 Mixed trophic impact (MTI) analysis of the 1960s’ Black Sea Ecopath model.

Red and blue colors signify negative and positive impacts, respectively.

The keystone species in the system was spiny dogfish, with an index value of −0.024, and was followed by the dolphins group with a keystone index value of −0.047 (Fig. 5). Spiny dogfish also had a high relative total impact (1.0) on the food web, followed by dolphins (0.94). Zooplankton and phytoplankton also had the second (−0.11) and third (−0.15) highest keystone indices with relative total impact values of 0.92 and 0.89, respectively. The fifth species with the highest keystone index value (−0.19) was the adult anchovy group, with a relative total impact of 0.8.

Figure 5 The keystone index and the relative total impact values of the functional groups in the 1960s’ Black Sea Ecopath model.

Time dynamic model skill

The results of the fitting procedure for the temporal model are listed in Table S3. The model with the lowest AICc score (83.1) was selected as the best fit model. The comparison of the time series of the simulated biomasses and catches and the reference data is provided in Figs. S3 and S4, respectively.

The simulated biomasses and catches of the state variables in the model were assessed using Taylor diagrams (Fig. S5). All of the root mean square deviations (RMSD) of the simulated model results were lower than 1.5 for biomasses and 2.0 for catches. Considering RMSD, the time-dynamic model performed well in reproducing the temporal dynamics of biomasses and catches of all fish species, zooplankton and phytoplankton, except sprat and shad. However, in contrast to the shad, the correlation between the simulated sprat catches and the reference catch values was high (~0.85). The model had the best skill in reproducing anchovy catches with a correlation coefficient of ~0.8 and RMSD distance close to 0.5. Considering biomasses, the model biases were low (<25%) for anchovy and zooplankton; moderate (25–50%) for sprat, horse mackerel, shad, Atlantic mackerel and turbot; and high (50–100%) for shad, whiting and dogfish. Considering catches, the model biases were low for anchovy, horse mackerel, shad, whiting, Atlantic mackerel, turbot, Atlantic bonito, and bluefish; moderate for dogfish; and high for sprat.

The regime shift analysis (1960–1999)

The STARS analysis detected shifts in all synthetic ecological and network indicators (Table 3, Fig. 6). The macro-ecosystem indicator TST showed four regimes with upward shifts in 1977, 1981 and a downward shift in 1994. Two upward shifts in 1970 and 1977, and one downward shift in 1983 were detected in structural indicator A/C. Similar shifts were observed in O/C, another structural indicator, but in opposite directions, and in addition, there was 1-year delay (1978) in the second shift, and additionally a fourth upward shift was detected in 1996.

Table 3 Timing and the directions (in parenthesis) of the shifts in the time series of ecological network indicators detected by the STARS algorithm.

Indicator	1960s	1970s	1980s	1990s	
Total system throughput (TST)		1977 (+)	1981 (+)	1994 (−)	
A/C		1970 (+), 1977 (+)	1983 (−)		
O/C		1970 (−), 1978 (−)	1983 (+)	1996 (+)	
Predatory cycling index (PCI)			1989 (−)	1996 (+)	
Finn’s cycling index (FCI)	1968 (−)	1973 (−), 1977 (−)		1996 (+)	
Finn’s mean path length (PL)	1965 (−), 1969 (−)	1976 (−)		1994 (+)	
Primary production (PP)		1977 (+)		1994 (−)	
Biomass		1977 (+)		1995 (−)	
Catch		1972 (+), 1979 (+)	1984 (−)		
Proportion of flows to detritus		1970 (+), 1978 (+)		1996 (−)	
Mean trophic level of the catch (mTLc)	1964 (−)	1971 (−)			
Diversity			1984 (−)	1996 (+)	
Fishing in balance index (FiB)	1967 (+)	1972 (+), 1979 (+)	1984 (−)		
Loss of production index (L-index)		1979 (+)	1984 (−)	1996 (+)	
Primary production required to sustain catches (PPRc)	1964 (−)	1979 (+)	1989 (−)		
The probability of sustainable fishing (Psust)				1997 (−)	
Ratio of fodder zooplankton to total zooplankton		1973 (−)	1981 (−)	1997 (+)	
Ratio of predatory fish to forage fish biomass	1965 (−)		1980 (+)	1990 (−)	
Ratio of demersal fish to pelagic fish biomass	1965 (−)		1984 (+)		
Ratio of fish to invertebrate biomass	1967 (−)	1973 (−)			
Note:

Positive signs denote an upward (increasing) shift, and negative signs denote a downward (decreasing) shift in the mean value of the indicators.

Figure 6 Results of the STARS regime shift analysis on the anomalies of the synthetic ecological indicators from the Ecosim network analysis between 1960–1999.

Bars represent time series of indicators’ anomaly values; blue solid lines represent different regimes detected by the regime shift analysis. TST, total system throughput; PCI, predatory cycling index; FCI, Finn’s cycling index; PL, path length; PP, primary production; PFD, proportion of flows to detritus; mTLc, mean trophic level of the catch; FiB, fishing in balance index; L-index, loss of production index; PPRc, primary production required to sustain catches; Psust, the probability of sustainable fishing index.

The STARS analysis also detected shifts in the food web (ecosystem function) indicators. The FCI showed three downward shifts in 1968, 1973 and 1977, as an indicator of decreasing material cycling in the food web, and an upward shift in 1996. Due to this decreasing cycling, the PL also showed three downward shifts; however, with different timings in 1965, 1969 and 1976, and one upward shift in 1994. The PFD showed upward shifts in 1970 and 1978, and a final downward shift in 1996, supporting the decreasing cycling detected in FCI and PL, and indicated a short-circuit of flows back to detritus instead of being transferred to higher trophic levels in the food web. PCI, which also shows cycling in the food web but excludes the flows through detritus, showed a downward shift in 1989 and an upward shift in 1996. Shannon’s diversity index showed a downward shift in 1984 and an upward shift in 1996, indicating degraded and alleviated conditions, respectively, in the proportion of organisms in the Black Sea food web.

Shifts in fishery-related indicators, i.e., catch, mTLc, FiB, PPRc, L-index, and Psust, were also detected by the STARS algorithm. STARS detected upward shifts in catch in 1972 and 1979 and a downward shift in 1984. Similarly, upward shifts were observed in the FiB in 1967, 1972 and 1979, and a final downward shift was detected in 1984. The PPRc index, an indicator of the fisheries footprint on the ecosystem, showed a downward shift in 1964, an upward shift in 1979 and a final downward shift in 1989. The mTLc showed downward shifts in 1964 and 1971, indicating a fishing down the food web effect, and STARS did not detect any significant shifts thereafter. The L-index showed an upward shift in 1979, a downward shift in 1984 and finally an upward shift in 1996. The probability of the sustainable fishing index, Psust, showed only one downward shift in 1997.

The ratio of fodder zooplankton biomass to total zooplankton biomass, an indicator of an increase in opportunistic species such as jellyfish, showed two downward shifts in 1973 and 1981 and a final upward shift in 1996. Similarly, the ratio of fish biomass to invertebrate biomass showed two downward shifts in 1967 and 1973. STARS detected a downward shift in 1965, an upward shift in 1980 and a final downward shift in 1990 in the ratio of predatory fish biomass to forage fish biomass. The ratio of demersal fish biomass to forage fish biomass showed a downward shift in 1965 and an upward shift in 1984. STARS analysis detected two significant shifts in the system’s primary production and biomass, upward shifts for both indicators in 1977 and downward shifts in 1994 and 1995, respectively.

Two components were retained in the PCA analysis based on the empirical Kaiser criterion (Fig. S6). The first two principal components of the PCA explained 51.8% (PC1) and 21.6% (PC2) of the variability in the time series of the indicator anomalies (Fig. 7). PC1 mainly represented the variability in the food web-related indicators, i.e., PFD, FCI, PL, and the ratio of fodder zooplankton biomass to total zooplankton biomass; fisheries-based indicators, i.e., the ratio of fish biomass to invertebrate biomass and FiB, and macro-ecosystem indicators, i.e., TST, PP and biomass. PC2 represented the variability in fishery-related indicators, i.e., the L-index and Psust, and function-related indicator, PCI. According to the PCA groupings, the 1960s ecosystem was characterized by mTLc, Shannon’s diversity index, the ratio of fodder zooplankton biomass to total zooplankton biomass, FCI, PL, the ratio of fish biomass to invertebrate biomass and O/C. The 1980s was characterized by A/C, catch, PFD, TST, biomass, PP and FiB. The 1970s and the 1990s ecosystems were similarly grouped and characterized by the A/C, O/C and FiB indicators.

Figure 7 Results of the principal component analysis of ecological indicators from Ecosim time-dynamic simulation showing the first (PC1) and second (PC2) standardized principal components and the factor loadings of the indicators.

Groups denote the decadal periods of the 1960s, 1970s, 1980s and 1990s in the 40-year simulation. TST, total system throughput; PCI, predatory cycling index; FCI, Finn’s cycling index; PL, path length; PP, primary production; PFD, proportion of flows to detritus; mTLc, mean trophic level of the catch; FiB, fishing in balance index; L-index, loss of production index; PPRc, primary production required to sustain catches; Psust, the probability of sustainable fishing index.

The time series of the L-index and its reference levels that correspond to the 50% (L50%) and 75% (L75%) probabilities of sustainable fishing showed that the probability of sustainable fishing (Psust) oscillated between 50% and 75% throughout the 40-year period in the Black Sea, with occasional decreases below the 50% probability of sustainable fishing during the 1960s, 1980s and the late 1990s (Fig. 8). The average ecosystem-based maximum sustainable catches corresponding to a 50% probability of sustainable fishing (EMSC50%) were calculated as 264 ± 32 kilotonnes in the 1960s, 504 ± 179 kilotonnes in the 1970s, 1,108 ± 262 kilotonnes in the 1980s, and 702 ± 121 kilotonnes in the 1990s. The average ecosystem-based maximum sustainable catches corresponding to the 75% probability of sustainable fishing (EMSC75%) were calculated as 103 ± 12 kilotonnes in the 1960s, 196 ± 69 kilotonnes in the 1970s, 431 ± 102 kilotonnes in the 1980s and 273 ± 47 kilotonnes in the 1990s.

Figure 8 Time series of L-index calculated by the Ecosim network analysis routine between 1960-1999 and corresponding EMSC values.

The dash-dotted and dashed lines denote corresponding L-index values for 50% and 75% probability of being sustainably fished, respectively. Right, time series of catches (solid black line) and corresponding ecosystem-based maximum sustainable catches (EMSC) for the 50% (dash-dotted line) and 75% (dashed line) probability of being sustainably fished, respectively.

Discussion

Our study showed that: (i) the 1960s’ ecosystem was characterized by top-down control exerted primarily by spiny dogfish and dolphins, as shown by the keystoneness analysis, and strong negative impacts of fishing fleets on these two top predators and predatory fishes, as indicated by the MTI analysis; (ii) the regime shifts in the Black Sea ecosystem were reflected by the time series of ecological indicators; and (iii) four different regimes were experienced in the Black Sea between 1960 and 1999. The first regime (circa 1960–1968) was characterized by a resilient and healthy ecosystem with low levels of TST, biomass, catch, PFD and A/C, and high levels of O/C, FCI, PL and mTLc, but the ecosystem was on the brink of changes due to overexploitation of top predators by fisheries, as indicated by the negative MTI on apex predators and high PPRc index. The second transitory regime (circa 1969–1976) was mainly characterized by elevated eutrophic conditions with increasing values of TST, biomass, catch and A/C, and decreasing values of O/C, FCI, PL and mTLc, as also indicated by the increasing PP values. During the second regime, opportunistic zooplankton groups established their populations, and a significant portion of the flows in the food web was short-circuited back to detritus, as indicated by the increase in the PFD, due to the low utilization and low ecotrophic efficiencies of these opportunistic groups in the food web. The third regime (circa 1977–1989/1994) was characterized by the most stressed conditions in the ecosystem with the highest values of TST, biomass, catch and PP, and the lowest values of FCI and PL, and with a high footprint of fisheries, as indicated by the L-index and PPRc, and decreasing diversity. The fourth and final regime (circa 1994–1999) was characterized by conditions similar to the 1970s’ transitory regime; however, in the reverse direction with decreases in TST, biomass, catch, PPRc, L-index, A/C, and slight increases in O/C, FCI, PL and mTLc, indicating a limited alleviation of adverse conditions experienced in the third regime.

Model skill

The 1950s and the 1960s in the Black Sea lacked systematic scientific studies, although a good deal of knowledge was collated afterwards (e.g., Ivanov & Beverton, 1985; Prodanov et al., 1997; BSC, 2008). Therefore, the data quality of the initial conditions and input parameters could be hindered by insufficient stock assessments and fisheries biological data for certain modelled periods. However, according to Morissette (2007), the pedigree index of 48% of the Ecopath models published ranged between 0.4–0.599, included between 20 and 40 functional groups, and only 10% of the models had a pedigree index above 0.6. Therefore, the pedigree index of the Black Sea Ecopath model was considered satisfactory, indicating that the model had a moderate degree of data quality.

The PREBAL analysis of the final balanced Ecopath model indicated possibly overestimated P/B and Q/B ratios for jellyfish species, i.e., B. ovata and M. leidyi, and zooplankton and phytoplankton, considering their respective trophic levels. Jellyfish can attain significantly higher productivity rates (Pitt et al., 2013) and can be considered outliers in this respect. The Ecopath model of the Black Sea was balanced by capitalizing on the pedigree values of the data sources for the parameters and initial conditions and calibrating from the most uncertain to the least uncertain parameter/initial condition values. Generally, the biomasses were the values with the highest uncertainty, considering that the majority of stock assessment studies and reliable catch statistics dated back to the early 1970s. Therefore, the biomasses of whiting, horse mackerel, sprat, anchovy, zooplankton, and phytoplankton were overestimated as a result of balancing the model.

In the temporal Ecosim model of the Black Sea, the correlations of simulated catches and biomasses to the statistical catches and predicted biomasses by statistical stock assessments varied; however, the RMSDs were low, the simulated time series of catches and biomasses matched the trajectory of the reference values, and the biases were mostly low or moderate in the validated 40-year Ecosim model of the Black Sea (Fig. S5). The model could reproduce the decadal changes in the time series of statistical catches and the statistical stock assessment predicted biomasses as well as the anchovy stock’s collapse in 1989 (Figs. S3 and S4). It should be noted that the reference data used to validate the temporal model had high uncertainty because of the questionnaire-based collection of the statistical catches in the Black Sea, statistical stock assessment-based predictions that capitalized mostly on these statistical catches, and/or limited survey data, or extrapolation of catch-at-age information from other countries to the catches of riparian countries where no such data were available. Therefore, the model can be considered successful in reproducing the food web dynamics of the Black Sea from 1960 to 1999.

The state of the early 1960s’ Black Sea ecosystem

The early 1960s’ Black Sea ecosystem was highly resilient (73.5% O/C) to stressors and the transfer efficiency in the food web was high (Table 2). Shannon’s diversity index, which has an upper bound of approximately three considering the number of species represented in the model, indicated high diversity during this period, and the mTLco was also high. Net system production was above unity and showed that the Black Sea was in a developmental stage sensu Odum (1969). However, the gross production-to-community respiration ratio and the primary production-to-total respiration ratio were above but close to unity. Therefore, we hypothesized that the Black Sea was at a developmental stage in the 1960s; however, it was close to maturity. The SOI was low compared to the 2000s’ conditions in the Mediterranean Sea (Piroddi et al., 2015; Coll et al., 2016) and the early 1990s’ conditions in the North Sea (Mackinson, 2014) and higher than the 1974’s conditions in the Baltic Sea (Tomczak et al., 2012), and showed specialized feeding interactions in the food web. Although the FCI was high, the PCI was quite low compared to that of the Mediterranean Sea (Piroddi et al., 2015); however, it was higher than that of the 1974’s Baltic Sea ecosystem (Tomczak et al., 2012). In terms of fisheries footprint on the ecosystem, the L-index and the low PPRc value, which was slightly greater than 5%, indicated a sustainably fished ecosystem with a probability of approximately 89% compared to the 1963’s North Sea (62.6% PPRc and 6.4% Psust) or the 1974’s Baltic Sea (36.3% PPRc and %0 Psust) (Libralato et al., 2008). The low L-index value was due to fishing, which mainly exploited higher-trophic-level species. However, the L-index considers the whole fisheries catch and system’s productivity and does not mean that sustainable fishing occurs at individual stocks’ levels; therefore, it should be complemented by stock-by-stock evaluations (Libralato et al., 2008). An interesting finding was that the gross efficiency of the Black Sea in the 1960s was comparable to that of the ecosystem-overfished Baltic Sea in the 1970s (Libralato et al., 2008; Tomczak et al., 2012), possibly because the Baltic Sea model was not forced by primary production anomalies.

The keystoneness analysis of the quasi-pristine 1960s’ mass-balance model of the Black Sea revealed that the ecosystem was under top-down (consumer/predator) control by apex predators, i.e., marine mammals and spiny dogfish (Fig. 5). The influential roles of zooplankton and phytoplankton were also significant, because these two groups followed the two predators in terms of keystoneness. Top-down control during the 1960s in the Black Sea food web was also supported by previous studies; however, the consumer control was exerted by Atlantic mackerel, Atlantic bonito, bluefish and/or marine mammals (Daskalov et al., 2007, 2017; Oguz, 2007; Llope et al., 2011). Contrary to previous findings, in our study, consumer control in the 1960s’ food web was exerted primarily by spiny dogfish and then marine mammals, and the impacts of Atlantic mackerel, Atlantic bonito, and bluefish were relatively low in terms of their structuring roles and relative total impacts on the ecosystem (Fig. 5). The mixed trophic impact analysis indicated that predator fishes, i.e., Atlantic bonito, bluefish and Atlantic mackerel, were under top-down regulation by marine mammals. The strong negative impacts of trawlers on turbot and spiny dogfish were alarming because spiny dogfish was one of the apex predators in the ecosystem with the highest keystoneness and relative total impact, and a small change in the biomass of spiny dogfish would be prone to significant changes in the ecosystem. Furthermore, the dolphins group was negatively impacted by seiners due to hunting, and after the whaling ban, intentional killings. Overall, the two most influential species/groups in the food web of the 1960s’ Black Sea were negatively impacted by fisheries. The negative impacts of fisheries on three predatory fish species, i.e., Atlantic bonito, Atlantic mackerel and bluefish, were also significant. Considering the entire food web, the transfer efficiencies of energy through the trophic levels in the food web were high up to TL IV and close to the ecologically theoretical value of 10% at TL V. This indicated an efficiently functioning food web, although higher-trophic-level species/groups were overexploited by fisheries.

The regime shifts in the Black Sea were reflected by the synthetic ecological indicators calculated by the ecological network analysis for the first time. Overall, the Black Sea was characterized by four distinct regimes. Although the detected timings of shifts were different for different indicator groups (macro-ecosystem, food web, and fishing indicators), the changes in their time series were similar. We hypothesized that the first regime shift was instigated by fisheries overexploitation of higher-trophic-level species, as indicated by keystoneness and MTI analysis for the 1960s and the temporal changes in the L-index and Psust values. The second regime was driven by increasing productivity due to eutrophication, as indicated by the PP. The third regime was realized by intensified eutrophication due to a continuous increase in the primary productivity and a related increase in the opportunistic (r-selected) and trophic dead-end species, as shown by the decrease in the ratio of fodder zooplankton biomass to total zooplankton biomass. The final regime was instigated by fish stocks’ collapse due to overexploitation and trophic competition, and settled with alleviated eutrophic conditions, as shown by the PP.

From its quasi-pristine stage in the 1960s, the Black Sea ecosystem deteriorated continuously until the end of the 1980s. The mTLc decreased significantly until the 1970s, and the effect of fishing down the food web (Pauly et al., 1998) was evident, as also shown by the L-index and PPRc. Similar conclusions were reached by long-term time-series data analysis of biomasses and catches (Daskalov et al., 2007) and statistical modelling practices capitalizing on time series of field data (Llope et al., 2011; Daskalov et al., 2017), which underlined the removal of the stabilizing effect of apex predators in the Black Sea. Hence, a decrease in the ecosystem’s resilience occurred because of overexploitation by fisheries. However, the proportion of flows to detritus was low after the removal of apex predators until the onset of eutrophication. This indicated that, although decreased, the resilience of the Black Sea ecosystem was preserved to a certain extent until the PFD increased because of the increased biomasses of trophic dead-end organisms such as jellyfishes and heterotrophic dinoflagellates by the mid-1970s. The regime shift detected in the ratio of fodder zooplankton biomass to total zooplankton biomass after the onset of the 1970s also supported this conclusion. This change also coincided with the increased primary productivity due to eutrophic conditions (Oguz, Akoglu & Salihoglu, 2012). The changes in the FCI and PL indicators also aligned with the changes in the mTLc and PP indicators to a certain extent; however, sometimes with different time frames. Although somewhat contrasting with the results of the mass-balance Ecopath model, the high PPRc in the first half of the 1960s indicated that the fishery had a significant ecological footprint due to the exploitation of high-trophic-level species during this period, as also indicated by Llope et al. (2011) and Daskalov et al. (2017). After the mid-1960s, the PPRc remained at low levels until the onset of the 1980s, when the fishing fleet’s capacity (FiB in Fig. 6) and catches reached their maxima. The increasing FiB index since the mid-1960s and the increasing catches until the mid-1980s reflected the increased capacity of the fleet and its enormous catches. However, the carrying capacity of the Black Sea ecosystem increased due to elevated primary productivity levels caused by intense eutrophication, as shown by the changes in the TST, PP and the total biomass supported by the system. Therefore, the L-index remained at levels corresponding to the 50% and 75% probability of sustainable fishing in the ecosystem. The removal of higher-trophic-level species relieved the predatory (top-down) control on the medium- and lower-trophic-level fish species and, along with the increased carrying capacity due to bottom-up (resource-driven) mechanisms, large amounts of biomasses and fisheries catches were possible during the late 1970s and the early 1980s.

PCA analysis showed that the ecosystem conditions in the 1960s and the 1980s differentiated in adverse directions based on the time series of ecological indicators and could be considered “ecological opposites”. The 1960s was characterized by higher values of mTLc, diversity, FCI, PL, O/C, and higher ratios of fodder to total zooplankton and fish to invertebrate biomasses, whereas the 1980s was characterized by higher values of A/C, catch, PFD, TST, biomass, PP and FiB indicators. The 1970s and the 1990s were between these two opposite characteristics of the 1960s and the 1980s, and could be considered transitional periods, or moderate regimes/states based on productivity and ecosystem structure and functioning. The settling changes during the 1960s, the gradual overexploitation of higher-trophic-level species, as indicated by mTLc, and the ratio of predatory fish biomass to forage fish biomass differentiated the early 1960s from the rest of the decade. The 1980s was grouped closer together than the 1960s, and the 1970s and the 1990s were grouped between these two decadal extremes of the Black Sea as they were an amalgamation of the characteristics of the two extremes.

The average decadal fishery catches in the Black Sea were 249, 460, 919, and 491 kilotonnes in the 1960s, 1970s, 1980s, and the 1990s, respectively (Pauly, Zeller & Palomares, 2020). Considering the calculated EMSC50% values, the average decadal catches were lower than, but close to, the calculated catch values, and the variability was high. Considering the calculated EMSC75% values, the average decadal catches were much higher. Therefore, the time series of L-index values and the corresponding EMSC values indicated that unsustainable fisheries in the Black Sea were highly probable during the 40 years investigated in this study, which was also supported by another study that relied on the analysis of statistical catches for the period 1970–2010 (Tsikliras et al., 2015).

One exception could be the estimates of the temporal Black Sea model for the 1960s, as the mass-balance Black Sea model for the early 1960s indicated a low L-index value, contrary to that calculated by the temporal simulation. This discrepancy could be an artifact of the model fitting by alternating primary production anomalies, as inferred from the comparison of decadal catches and calculated EMSC values for the 50% and 75% probability of sustainable fishing; therefore, the fisheries could be sustainable with a high probability during the early 1960s. However, it should be noted that the L-index and EMSC calculations relied on the assessment of total catches in the system and the carrying capacity of the ecosystem as per the algorithm of the indices. Therefore, the sustainability of fisheries at the individual stocks’ level cannot be inferred from the analysis, and our findings based on the L-index and EMSC should be tentatively evaluated.

Limitations of the study and future considerations

Although we did not conduct a formal uncertainty analysis regarding the model inputs, it was possible to infer the degree of uncertainty in the model outputs based on data availability. The data required to parameterize the 1960s’ Black Sea Ecopath model relied on the historical literature. Most fish stock assessments related to the 1960s were conducted two decades or more after the period by collating various sources of data on the Black Sea. However, the systematic connectivity of the data was constrained by the limited presence of basin-wide harmonized efforts to collect standardized and comparable survey data in the Black Sea at the time. In addition, although we parameterized the Ecopath model using data from the 1960s of the Black Sea to the most possible extent, certain parameters for some species were lacking in this period, and we occasionally had to capitalize on data from earlier or later decades (Table S1). Further, stock assessments of important predatory fish species were completely missing then and are still missing as of now. These factors may have caused uncertainty in the mass-balance model results, as indicated by the pedigree value used for assessing the input data quality.

The lack of a time series of primary productivity measurements in the basin was another drawback of the modelling. The most comprehensive literature to date was by Yunev et al. (2002), who attempted to collate all literature information and inferred primary productivity estimates for the pre-satellite era in the Black Sea. However, his effort was at the level of calculating annual average primary productivity estimates on decadal scales. Therefore, we capitalized on these average primary productivity calculations and calibrated primary productivity anomalies in addition to trophic interaction parameters to fit the model to the observations of biomasses, catches and field-sampled data. Although this significantly increased the skill of the temporal model, it may have created a degree of uncertainty in the simulated primary productivity levels. The impact of this exercise could be reflected in the discrepancy between the L-index and Psust values calculated for the 1960s in the temporal model simulation and the Ecopath model of the 1960s’ Black Sea. However, this discrepancy did not hamper our results and related conclusions because we did not rely on a single or a few indicators in the study to assess the ecosystem structure and function but used a wide set of ecological indicators comparatively to develop an understanding of the historical changes in the Black Sea ecosystem. Future work should focus on improving the historical primary productivity estimates for the EwE model to decrease the source of uncertainty originating from phytoplankton dynamics, possibly by forcing the model by capitalizing on biogeochemical model products, if available.

Conclusions

Black Sea studies lacked holistic ecological assessments of its whole ecosystem and were limited to a certain part of the food web or used data analysis with statistical models that provided a useful but limited understanding of its historical changes. Furthermore, mechanistic models developed for the Black Sea were either static or lacked temporal validation. Therefore, we aimed to provide an ecosystem-based understanding of the historical changes in the Black Sea ecosystem for the first time with a validated long-term temporal model of the Black Sea ecosystem covering a 40-year period of its history in the second half of the 20th century and to develop the first holistic ecosystem approach to understand historical changes in the Black Sea using the fundamentals of ecology and related ecological indicators.

The mass-balance model results showed that the 1960s’ ecosystem was influenced by top predators, i.e., spiny dogfish and dolphins. The impact of fisheries on predatory fish species was already high, although fisheries were most likely at sustainable levels. The temporal model results and related regime shift analysis indicated that four regimes were experienced over the course of 40 years of history in the Black Sea between 1960 and 1999. The first regime of the 1960s was characterized by high degrees of material cycling and a rich biodiversity, with a high mean trophic level of catches. With the onset of the 1970s, the influence of top predators in the ecosystem abated, and the opportunistic zooplankton community was established as eutrophication intensified. Small pelagic fish species came into play during the decade as mediators of energy distribution in the food web. In the 1980s, eutrophication culminated in its peak, material cycling was at its lowest level, and the proportion of flows diverted back to detritus deprived the higher trophic levels of the great productive capacity of the system. This lasted until the collapse of the fishery in the late 1980s and alleviation of eutrophic conditions in the early 1990s, and a period similar to the early 1970s prevailed thereafter.

Overall, our results showed that anthropogenic disturbances such as eutrophication and overfishing were the main instigators of the different regimes in the Black Sea, and the adverse ecosystem conditions were exacerbated by interspecific competition in the food web. The immediate implementation of a basin-wide regulatory mechanism for human activities is crucial to prevent future catastrophic regime changes in the Black Sea, particularly under the increasing impact of climate change.

Supplemental Information

Supplemental Information 1 Data sources of the input parameters for the 1960s Black Sea Ecopath model.

Click here for additional data file.

Supplemental Information 2 Relative diet composition matrix for the 1960s Black Sea Ecopath model.

Click here for additional data file.

Supplemental Information 3 Results of the time series fitting by sum of squared deviations (SS) and Akaike Information Criterion corrected for small number of observations (AICc).

The best model is the one with the lowest AICc score.

Click here for additional data file.

Supplemental Information 4 Time series of fishing mortalities (F, y−1) and primary production anomalies used to force the model.

Note that the dolphins fishery halted due to basin-wide fishing ban in 1982.

Click here for additional data file.

Supplemental Information 5 PREBAL plots of biomasses, P/B, Q/B and P/Q for the modelled functional groups and species.

Bars show the values and black straight lines show the trend of the values. X axes show the functional groups and species sorted from high to low trophic level groups, and species sorted from high to low trophic levels.

Click here for additional data file.

Supplemental Information 6 Time series of simulated biomasses (lines) vs XSA stock assessment predicted biomass values for fish groups and in-situ data for zooplankton and phytoplankton (dots).

Click here for additional data file.

Supplemental Information 7 Time series of simulated catches (lines) vs statistical catches (dots).

Click here for additional data file.

Supplemental Information 8 Taylor diagrams showing temporal Black Sea model’s skill for biomasses and catches.

X and Y axes show standard deviations normalized in reference to the observations (REF on the X axis), the radial axis shows correlation coefficients of the model-simulated biomasses and catches to the XSA stock assessment and in-situ biomass and statistical catch values, and the dashed semi-circles show root mean square distance (RMSD) of the model-simulated values to the observations (REF). The skill of the model in reproducing the reference data increases towards the REF point on the X axis. Relative biases are shown with circles and triangles and the magnitude of the biases are proportional to the size of the shapes.

Click here for additional data file.

Supplemental Information 9 Eigenvalues and corresponding reference values of the principal components in the PCA.

Principal components with eigenvalues that are greater than their corresponding reference values are retained considering the empirical Kaiser criterion.

Click here for additional data file.

Supplemental Information 10 Black Sea Ecopath Model Input Parameters.

Click here for additional data file.

Supplemental Information 11 Black Sea Ecopath Model Relative Diet Composition Matrix.

Click here for additional data file.

Supplemental Information 12 Black Sea Ecosim Model Time Series of Forcing Functions Data.

Click here for additional data file.

Supplemental Information 13 Black Sea Ecosim Model Time Series of Ecological Indicators Data for Regime Shift Analysis.

Click here for additional data file.

Additional Information and Declarations

Competing Interests

Author Contributions

Data Availability

The author declares that they have no competing interests.

Ekin Akoglu conceived and designed the experiments, performed the experiments, analyzed the data, prepared figures and/or tables, authored or reviewed drafts of the article, and approved the final draft.

The following information was supplied regarding data availability:

The model input parameters and initial conditions as well as data used to validate the model simulation are available in the Supplemental Files.

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
