# Peer review of "Ecological indicators reveal historical regime shifts in the Black Sea ecosystem"

_PeerJ, doi:10.7717/peerj.15649_

## Round 0.1 · original submission · Minor Revisions

All referees agree that it is an original and well-written manuscript, but there are some minor and customary revisions that they requested before the final acceptance of the manuscript.

·

Basic reporting

Clear and unambiguous, professional English used throughout.

Experimental design

Research question well defined, relevant & meaningful. It is stated how research fills an identified knowledge gap.

Validity of the findings

no comment

Additional comments

Dear Autor,
I read the article you sent, even though it's not my area of expertise. The article is very well structured and well written. It offers great information from the table to the figures. I think it can be accepted into your journal with its writing, grammar, and editing.
Kind regards

Reviewer 2 ·

Basic reporting

The manuscript is clearly written in professional, unambiguous language. It will be better if it can be shorten and more condensed. The paper offers an new insight on understanding the Black Sea ecosystem state historically and the author also put forward practical measures on cross-national management efforts, which both are important for ecosystem management.

Experimental design

Line 265, the data of phytoplankton and zooplankton is from Ivanov & Beverton (1985), but the results is between 1960-1999. So where does the data of phytoplankton and zooplankton between 1960-1999 come from? Is it a prediction data? Please clearly describe the source of the data used.

Validity of the findings

no comments

Additional comments

1. Line 246 km2, where the 2 should be in superscript.
2. The input parameters are fundamental to the model output, many parameters are obtained from published studies, it could be used. But the author need to state parameters were from different period, and how it will affect the output of the model and results and conclusion.
3. Line 509 to line 513 , these sentences are about methods, can it be include in the section materials and methods?
4. The regime shift analysis (1960-1999), line 532 to 561, we can see that regime shift were observed by different analysis/ indicators, the shifts occurred in different year, and some were downward, while some were upward. Some were opposite in same year, such as TST upwared shifit in 1970 to 1977, but FCI down shifts in 1977. The author should explain more on how to understand these complicated or even opposite shifts.
5. Line 573 to line 574, it will be better if the author can add the contribution of principal components or Principal component gravel diagram, which is helpful to understand if the first two principal components PC1 and PC2 are sufficient to explain the results.
6. Line 827 , The section Conclusion should be abbreviated and more condensed, it seems a little bit longer.

·

Basic reporting

The manuscript #83613 from PeerJ entitled “ Ecological indicators reveal historical regime shifts in the Black Sea ecosystem” uses the EwE software to develop an Ecopath model for the Black Sea during the 1960s and an Ecosim model extending form 1960 until 1999 validated with biomass and catch time-series data. Overall it is a complete effort implemented by the author, with a well-written manuscript, describing the food web of the Black Sea during important periods.

Experimental design

Regarding calibration of Ecosim model and fitting to time series it is not clear from the text and Table 4 if the author used both time series of primary productivity and the primary production anomaly of the software with spline points in order to drive the model. Also it would be beneficial to identify the number of vulnerabilities as the trophic forcing factor and the number of spline points as the anomaly forcing function through the fitting process if such a driving factor was used.

Although the author mentions the uncertainty issue in the results and discussion sections it is not clear in the text if the Monte Carlo routine was used, which is the incorporated software’s tool for addressing uncertainty in Ecosim models. Since uncertainty is addressed in the manuscript it would be optimal for the author to provide some details regarding this issue.

Validity of the findings

The validity of the findings is provided throughout the manuscript with well-endorsed results. Also it is important that the original Ecopath configuration of the 1960s in the Black Sea (Akoglu et al. 2014) was updated with the inclusion of more compartments of the food web.

Additional comments

Minor comments:
1. L.83: To add Shalovenkov N (2019) Chapter 31 Alien Species Invasion: Case Study of the Black Sea. Coasts and Estuaries 547-568.
2. L.114: ‘which had a stabilizing role in the ecosystem by fishing’. To rephrase to: which had a stabilizing role in the ecosystem, through fisheries.
3. L.150: To correct the numbering from ii) to iii).
4. L.171: To replace alien invasive species with non-indigenous species (NIS). I believe it is better since not every alien species is an invasive one.
5. L.224: To rephrase to: [(Macartney) Kofoid & Swezy, 1921].
6. L.234-235: To rephrase to: values close to zero and forced to be equal to zero until their period of introduction.
7. L.279: To italicize the parameters’ abbreviations (also throughout the rest of the text where needed).
8. L.307: To replace ‘were’ with ‘was’.
9. L.342: To replace ‘ecosystem’ with ‘ecosystems’.
10. L.599: To insert a gap in ‘1960secosystem’.
11. L.619: To delete one ‘the’.
12. L.708-719: The part here is more appropriate in the results section. However, this information has already been provided in L.603-616. It should be rephrased or deleted.
13. L.733: To delete one ‘(Pauly et al., 1998)’.

---

## Round 0.2 · accepted · Accept

Both referees have underlined the high quality of the paper and the efforts spent by the authors to address their suggestions; therefore the paper can be accepted in its present form.

Reviewer 2 ·

Basic reporting

No more suggestions.

Experimental design

arleady clearly stated as comments and sugesstions before mentioned.

Validity of the findings

already clearly stated.

·

Basic reporting

As i have stated the manuscript is a complete effort of the Ecopath and Ecosim modelling approach in the Black Sea and well-written, The new structure of the manuscript following the instructions from all reviewers improved the original version.

Experimental design

My remarks about calibration of the Ecosim model and uncertainty were resolved by the author.

Validity of the findings

The validity of the findings are supported throughout the revised version of the manuscript.

Additional comments

All the minor comments were addressed and properly resolved.